# Migraine as a Disease Associated with Dysbiosis and Possible Therapy with Fecal Microbiota Transplantation

**DOI:** 10.3390/microorganisms11082083

**Published:** 2023-08-14

**Authors:** Ágnes Kappéter, Dávid Sipos, Adorján Varga, Szabolcs Vigvári, Bernadett Halda-Kiss, Zoltán Péterfi

**Affiliations:** 11st Department of Internal Medicine, Department of Infectology, University of Pecs Clinical Centre, H7623 Pécs, Hungary; kappeter.agnes@pte.hu (Á.K.); sipos.david@pte.hu (D.S.); szabolcs.vigvari@gmail.com (S.V.); kiss.bernadett2@pte.hu (B.H.-K.); 2Department of Medical Microbiology and Immunology, University of Pecs Clinical Centre, H7624 Pécs, Hungary; adorjanvarga@ymail.com

**Keywords:** migraine, short-chain fatty acid, probiotic, fecal microbiota transplantation

## Abstract

Migraine is a painful neurological condition characterized by severe pain on one or both sides of the head. It may be linked to changes in the gut microbiota, which are influenced by antibiotic use and other factors. Dysbiosis, which develops and persists as a result of earlier antibiotic therapy, changes the composition of the intestinal flora, and can lead to the development of various diseases such as metabolic disorders, obesity, hematological malignancies, neurological or behavioral disorders, and migraine. Metabolites produced by the gut microbiome have been shown to influence the gut–brain axis. The use of probiotics as a dietary supplement may reduce the number and severity of migraine episodes. Dietary strategies can affect the course of migraines and are a valuable tool for improving migraine management. With fecal microbiota transplantation, gut microbial restoration is more effective and more durable. Changes after fecal microbiota transplantation were studied in detail, and many data help us to interpret the successful interventions. The microbiological alteration of the gut microflora can lead to normalization of the inflammatory mediators, the serotonin pathway, and influence the frequency and intensity of migraine pain.

## 1. Introduction

Microbiota is a key factor in our survival and its existence is fundamental to human health. A lot of agents define the diversity in microbiome content among individuals, such as environment, nutrition, age, genes, infections, and antibiotic use. The largest numbers of the human microbiota live in the gut. The number of intestinal microorganisms is higher than the number of body cells. We have approximately 10^13^–10^14^ microbes in our gut. They affect a lot of systems and organs in the human body. One such organ is the brain. Recently, the intestinal microbiota has been regarded as an important regulator of the intestinal brain axis, the term which refers to a bidirectional link between the intestine and the brain [1,2,3].

The intestinal microflora, or microbiome, is a complex system that affects many of the body’s functions. In addition to digestion, the microbiota plays a role in immunomodulation, modulating the inflammatory processes, and the endocrine system, and plays a role in neurological and behavioral processes through the gut–brain axis. Its role has been verified in some oropharyngeal, or gastrointestinal tumors, as well as their antitumor effects. In recent years, the examination of the intestinal microbiota has become a hot topic, and more than 53,000 publications have appeared on the subject area. The next generation sequence analysis (NGS), the Human Microbiome Project, and the Metagenomic of the Human Intestinal Tract (MetaHIT) project provide a large amount of data about the composition of the microbiota. A link has been found between the composition of the intestinal microbiota and several diseases and symptoms, such as pain, cognitive function, and neurodegenerative diseases [1,2,3,4,5,6]. This link suggests not only that the gut microbiota composition can serve as the basis to make an early diagnosis of neurodegenerative and neurodevelopmental diseases but also that modifying the gut microbiome to influence the microbiome–gut–brain axis might present a therapeutic target for these diseases [7]. The composition, density, and diversity of our individual microbiomes are dynamic and affected by multiple factors. Few of these factors including lifestyle, diet, geography, genetics, immune system, history of infections, and a plethora of environmental components can be modified [7].

Through the gut–brain axis, the central nervous system can also affect the intestinal microenvironments by regulating intestinal movement, excretion, and mucosal immunity. Vagal nerve, tryptophan metabolites, and microbial products such short-chain fatty acids (SCFAs) or peptidoglycan are the main channels of communication between the intestinal microbiota and the brain [8]. External factors such as diet, lifestyle, infections, antibiotic therapy, and hormones also regulate the composition of the intestinal microflora. At the same time, bacteria also act on the nervous system’s neurotransmitters and neuromodulators as a response to the effects mentioned above. Such modulators are of bacterial origin, such as choline, tryptophan, short-chain fatty acids, or hormones such as ghrelin or leptin. Changes in the intestinal flora can sometimes become self-sustaining and lead to the development of chronic non-infectious diseases. When detected in time, changing the external factors can reverse the process. Changing diet and lifestyle is a slow and time-consuming, yet not fully understood way to restore intestinal flora. Fecal microbiota transplantation (FMT) can provide a quick and effective option. However, accepted indications, which are based on randomized clinical trials and supported by evidence, exist only for the treatment of recurrent infections caused by *Clostridiodes difficile*. The effectiveness of FMT is also being investigated in diseases affecting the intestinal tract and non-gastrointestinal diseases, such as neurological and psychiatric diseases. It can be assumed that restoring the composition of the intestinal flora can help in the treatment of these patients [9].

## 2. Causes of Dysbiosis

Dysbiosis is a condition where the intestinal flora is altered due to earlier antibiotic therapy or other factors. It can cause various diseases such as metabolic disorders, obesity, hematological malignancies, and neurological or behavioral disorders. Some studies have suggested that dysbiosis can affect the gut–brain axis and contribute to some neurological diseases, such as anxiety, depression, autism, Alzheimer’s disease, stroke, Parkinson’s disease, and migraine [10,11,12,13,14,15,16].

Diet is the most common cause of dysbiosis worldwide. The influence of a long-term diet on the intestinal microbiome is far-reaching. The Western diet has sped up changes in the human microbiome. Highly processed foods, animal products, and sugars alter the microbiome to cause intestinal inflammation [7].

It has been demonstrated that many environmental factors influence the microbiome. Lifestyle, socioeconomic factors such as alcohol dependence, smoking, and sedentary lifestyle may be related to microbial dysbiosis.

Medicines can affect the balance of the microbiome and modify the microbiome unfavorably. Antibiotics and proton pump inhibitors usually reduce bacterial diversity and increase the number of potential pathogens such as *Clostridiodides difficile*. Selective serotonin reuptake inhibitors affect intestinal motility and, like antibiotics, may reduce microbial diversity [7]. One prior observational cohort study that examined the safety and tolerability of various antibiotic groups discovered a connection between taking antibiotics and experiencing migraines. More than 11,000 patients were treated with each antibiotic. The number of patients with headache and migraine increased two weeks after receiving fluoroquinolone, macrolide, or cephalosporin medication [17]. These antibiotics are the most potent ones that can change the microbiome in the gut. By increasing TNF production in an animal model, antibiotic therapy prolongs the pain from nitroglycerin-induced migraines [8].

## 3. Dysbiosis and Migraine

Migraine is a neurological disorder that causes severe pain on one or both sides of the head. There is increasing evidence that the alteration of the intestinal microbiota balance plays a role not only in gastrointestinal functions and immune system maturation processes, but it can also affect migraine [7,16,18,19]. The composition of the microbiota appears to be associated with certain pains, such as spinal cord, visceral pain, or pain in irritable bowel syndrome (IBS), as well as migraines and sometimes headaches. After antibiotic treatment, the intestinal flora is damaged, leading to dysbiosis and resulting in changes in colon sensor and motor functions. This effect can also be measured using nociceptive markers such as CB2 and TLR7 [1,20,21,22,23,24].

Amaral et al. showed that inflammatory hypernociception was induced by lipopolysaccharide (LPS), TNF-α, and IL-1β, and the chemokine CXCL1 was reduced in germ-free mice; such result was induced by prostaglandins and dopamine [1,25,26]. A meta-analysis has shown that Helicobacter pylori infection is associated with migraine [27], with 45% of migraine sufferers having H. pylori infection, compared to 33% of healthy control people. *H. pylori* infection is thought to be related to a persistent chronic inflammatory state, producing inflammatory and vasoactive agents [27,28,29,30]. Randomized double-blind clinical trials have reported improvement in migraine after eradication therapy [23]. Recent studies have also shown that chronic migraine-like pain induced by colon dysbiosis is associated with elevated levels of TNF-α and other proinflammatory cytokines [1,10].

In a recent study in migraine patients versus healthy people, Kopchak et al. found significant changes in the quantitative composition of some resident microorganisms. *Alcaligenes* spp., *Clostridium coccoides*, *Clostridium propionicum*, *Eggerthella lenta*, *Pseudonocardia* spp., and *Rhodococcus* spp. were found in greater numbers in migraine patients compared to the healthy group. The number of fungi-like *Candida* spp. and *Micromycetes* spp. increased in the migraine patients relative to the control group [31].

### Direct and Indirect Evindences of the Role of Gut Microbiota in Migraine

Several studies have shown that the gut microbiota is crucial to the emergence and maintenance of migraine. The relationship between altered gut microbiota and the development of migraines has both direct and indirect evidence [32].

Direct evidence of the role of the gut microbiota in migraine: Nitroglycerin (NTG) is commonly used to model migraine in rats. Wen and colleagues discovered that the pro-duction of metabolites by the microbiota was significantly altered by the prolonged usage of NTG [33]. In a different investigation, Lanza et al. discovered that the short-chain fatty acid can lessen the hyperalgesia caused by NTG [32,34]. More than 100 older women’s feces were studied, and it was discovered that reduced bacterial diversity and less butyrate production directly correlate with migraine [32,35].

Indirect evidence: The gut microbiota is affected by various external factors, such as diet, probiotics, vitamins, and lifestyle. There is indirect evidence for the gut microbiota’s participation in migraine. The results of double-blinded or placebo-controlled randomized trials demonstrated that external factors could affect migraine frequency and intensity [32].

Metabolites and their role in migraine: Metabolites generated by the intestinal microbiome also have been shown to influence the intestinal brain axis. Short-chain fatty acids (SCFAs) consist of more than 95% acetate, butyrate, and propionate (usually with a 60:20:20 ratio) and smaller amounts of valerate, fomate, and caproate and the branched-chain fatty acids isobutyrate, 2-methyl butyrate, and iso-valerate. Relationships are dependent on the mixing of bacteria in the colon. These molecules have important physiological effects [7]. Butyrate and propionate have been studied in more detail, and from this, butyrate is regarded as the most important SCFA. Bacterial butyrate inhibits histone deacetylase and stimulates memory and synaptic plasticity [36,37,38]. Butyrate also influences the release of serotonin from intestinal enterochromaffin cells [39]. The implication of serotonin in migraine pathogenesis is well documented; low serotonin levels can dilate blood vessels and initiate a migraine. Migraine sufferers often report that the headaches stop after they have vomited, which stimulates intestinal motility and raises blood serotonin levels (serotonin migraine) [40]. Capabilities for butyrate production among dominant intestinal species have been studied. The most important species belong to the Firmicutes family, such as *Faecalibacterium prausnitzii*, *Coprococus* spp., *Roseburia* spp., *Lachnosiraceae* spp., *Clostridial* Clusters IV and XIVa, and *Eubacterium hallii*, which are all butyrate-producing bacteria [41,42,43].

Propionate protects the blood–brain barrier (BBB) from oxidative stress [44]. Propionate production is related to the presence of other species, such as *Bacteroides vulgatus, B. uniformis, Alistipes putredinis, Prevotella copri, Roseburia inulinivorans, Veilonella* spp., and *Akkermansia mucinophila*, which have a high capability to produce propionate [42,43,44]. In addition, short-chain fatty acids can affect neural inflammation by modulating the pro-duction of immune cells and cytokines [45]. Certain metabolites produced by bacteria can act as essential neuroactive molecules in the central nervous system. A few species of *Lactobacillii* and *Bifidobacterii* can produce acetylcholine and gamma-amino butyrate (GABA) neurotransmitters [46,47]. Other species, such as *Enterococcus, Streptococcus*, and *Escherichia coli*, can synthesize serotonin, dopamine, and norepinephrine [46,47]. Gut microbes can metabolize tryptophan as a precursor for the synthesis of kynurenine, quinolonate, indole, indole acetic acid, other indole derivates, tryptamine, and melatonin, limiting the availability of this essential amino acid for the host [48,49]. The metabolic pathways of tryptophan are most effective in five intestinal phylums: *Actinobacteria, Firmicutes, Bacteroidetes, Proteobacteria, and Fusobacteria*. In addition, five genera, namely *Clostridium, Burkholderia, Streptomyces, Pseudomonas*, and *Bacillus*, also have greater potential for the metabolism of tryptophan in the intestine [31]. These tryptophan metabolites can exert an opposite effect on neuroactivity. The kynureine and quinolonate production depletes the supply of tryptophan and tryptamine as the precursors of serotonin. The literature suggests a probable role of certain genera such as *Holdemania, Tyzzerella, Desulfovibrio, Yersinia, Bacillus, Clostridium, and Ruminococcus* in regulating serotonin release through tryptamine production, positively influencing the host neurophysiology, in contrast to the negative effect of kynurenine and quinolinate as discussed above [31]. The most important metabolites in preventions of migraine severity and frequency and their producers are summarized in Table 1.

Besides serotonin, the role of other neuropeptides in migraine headaches is unquestionable. Glutamate plays a key role as a neurotransmitter in the pathophysiology of migraine through central sensitization by stimulating the trigeminovascular system. The activation of trigeminovascular pain routes is designed to mediate some of the qualities of migraine pain by releasing neuropeptides, such as calcitonin gene-related peptide (CGRP) and pituitary adenylate cyclase-activating polypeptide (PACAP) [50]. Through the intestinal brain axis, inflammation and the response to oxidative stress in the enteric nervous system can be affected. This activation of the trigeminovascular system is thought to cause the release not only of glutamate but of other vasoactive sensory neuropeptides, such as substance P [40,51] and CGRP [52], which further dilate cerebral blood vessels and produce an inflammatory response causing pain [53]. Elevated levels of circulating CGRP have been detected during migraines [40,53,54,55]. CGRP inhibits gastric acid secretion and reduces food uptake [28], and its elevated levels in dysbiosis can be counterbalanced by probiotics.

After a comparison of the effects of the intestinal microbiome on the central nervous system and their role in neurotransmitter production, we may select certain microbes that have a positive effect; others have a negative effect on the development of migraine, which are summarized in Table 2.

## 4. How to Modify the Gut Microbiota

### 4.1. Dietary Influence of the Intestinal Brain Axis

Different dietary approaches have been suggested to influence the intestinal brain axis. Healthy diets such as the Mediterranean diet, a traditional Japanese diet, Dietary Approaches to Stop Hypertension (DASH), the Mediterranean–DASH intervention for neurodegenerative delay (MIND) diet, vegetarian diets, ketogenic diets, and so on are known to produce a more balanced, anti-inflammatory microbiome over time. Patients on a ketogenic diet report less frequent and milder migraine episodes. While dietary change may help in inflammatory bowel disease, celiac disease, and other gastrointestinal diseases, the ability of a dietary change to alter neurological conditions is less clear and has so far yielded various results (Table 3) [7].

In humans, the effects of prebiotics are less easily established and have often been inconclusive, perhaps because of the heterogeneity of the gut microbiome and differences in study design. Although prebiotics may encourage some bacterial groups to increase in numbers, they do not increase the diversity of microbiomes. In conclusion, dietary approaches are extremely difficult to determine due to difficulties in maintaining dietary strategy, establishing dietary compliance, and designing blind studies [7].

### 4.2. Probiotics

The use of probiotics as a dietary supplement may reduce the number and severity of migraine episodes [56]. Its hypothetical effect is on SCFA production, improving the integrity of the epithelium, and improving proinflammatory cytokine levels by the suppression of kappa factor B [56,57,58,59,60,61,62]. The results of human trials are mainly related to neurodegenerative diseases such as Alzheimer’s disease and autism spectrum disorder (Table 3) [7].

### 4.3. Carbohydrates

Dietary switching to carbohydrates with a low-glycemic index has been shown to increase SCFA levels [28]. In one randomized study, the effect of the drug used for migraine prevention was compared with carbohydrate intake and an almost equal decrease in the severity of seizures was observed after 90 days on the diet [28,63].

Conversely, a high-fat diet reduces the growth of SCFA-producing bacteria and therefore has a less protective effect (Table 3) [28].

### 4.4. Vitamins

The B2 vitamin, or riboflavin, is crucial for oxidative metabolism and may lessen migraine risk. According to randomized controlled trials, taking vitamin B2 can significantly reduce the number of migraine days by 1–3 per month. By boosting the amount of short-chain fatty acid makers, microbial variety, and richness, the riboflavin and other vitamin Bs influence the gut microbiome in a favorable way (Table 2) [64].

Vitamin D3 supplementation can affect the intestinal microflora. In one study, 8 weeks of vitamin D3 supplementation significantly reduced *Helicobacter* spp. quantity. At the same time, patients with head pain and migraines have a higher vitamin deficiency than patients without pain (Table 3) [28,65].

### 4.5. Other

Magnesium insufficiency may affect the NMDA receptor blockade, calcium channel, glutamate, and NO activity, as well as the affinity of the serotonin receptor. Only a few studies have demonstrated a reduction in the length and severity of migraine headaches. In dysbiosis, magnesium serves as an adjuvant therapy (Table 3) [66,67].

Connected to the etiology of migraines are the n-3 and n-6 fatty acids. They act as a source of numerous oxylipins, which are bioactive lipid mediators and are regulators of pain. Strong antinociceptives include eicosapentaenoic acid (EPA) and docosahexaenoic acid (DHA). Ramsden et al. noticed a clinically relevant but not statistically significant decline in headache impact test (HIT-6) scores in one randomized clinical trial when compared to the control group. The oxylipin profile in healthy and obese rats is altered by gut microbiota dysbiosis, as shown by animal research, and there are several strong associations between various bacterial taxa and eicosanoids (Table 3) [68,69].

### 4.6. Fecal Microbiota Transplantation

Significant efficacy of FMT in recurrent *Clostridioides difficile* infection has been demonstrated by a large number of randomized controlled studies, systematic reviews, and meta-analyses, even though the mechanisms underlying its effectiveness are not fully understood. There are several concerns about its use, such as the unpleasantness of the procedure, the invasiveness of the intervention, the small but important risk of transmission of infections, and the difficulty of choosing the right donor [70]. The Food and Drug Administration (FDA) draws attention to potential serious risks, such as the possible transmission of multi-resistant pathogens [71]. This risk can be reduced by proper donor screening, or transplantation with bacteria-free filtrates, which Péterfi and his team found to be sufficiently effective, but these methods cannot be used to prevent the transmission of viruses. Using FMT, it is possible to transfer currently unknown microbes that may later be responsible for the development of chronic diseases. There are positive results in current clinical trials about FMT treatment in depression [72], but no clinical trials can be found on the clinicaltrials.gov website for migraine, nor can FMTs for migraine treatment be encountered. However, the triggers, and the possibility of changing them, give way to trying to treat recurrent or frequent migraine attacks with FMT. To support this, let us see what we know about fecal transplantation (Table 3).

## 5. Fecal Microbiota Transplantation and Their Effects

### 5.1. Donor Screening Criteria

When choosing a donor, it is especially important to exclude transmissible infectious diseases and multidrug-resistant (MDR) colonization during screening. The donor should have a normal stool habit and should not suffer from cancer or chronic metabolic disease. After a health check (including the previous six months of medication, travel, lifestyle, chronic disease, and stool habits), donor screening should include general blood collection and the exclusion of infectious agents that can be transmitted through feces. This should include screening for HAV, HBV, HCV, HEV, EBV, CMV, HIV, lues, toxoplasma, fecal parasites, bacteria, and the *C. difficile* toxin and antigen. Anal sampling should also be performed to exclude MDR-colonizing bacteria [72,73]. The COVID-19 pandemic also highlights the potential transmission of SARS-CoV-2. Given that commercially available tests are not routinely suitable for detecting the virus in feces and because the virus may be excreted for a long time after recovery, great caution and consideration is needed. Ianiro and colleagues have compiled a multi-step protocol to prevent the transmission of SARS-CoV-2 viruses with great certainty [74,75].

### 5.2. Transplantation Methods

Several methods have been tried in recent years to deliver donor stool to the recipient. Several studies and meta-analyses have compared the effectiveness of fresh and frozen feces and found no significant differences [76,77]. Vigvári and colleagues found that the preparation of lyophilized stool was as effective as fresh stool [78,79,80] and that the mode of administration through the nasogastric or nasojejunal tube was also similar [76,77]. Therefore, we can use fresh stool, as well as prepared stool, which can be frozen (at −80 °C) or freeze-dried. The fresh stool should be used within 6 h; frozen stool can be stored for about 6 months, and freeze-dried stool can be stored under appropriate conditions, theoretically indefinitely [8]. Lower gastrointestinal intake can be conducted in the form of an enema or by colonoscopy [8,71], while upper gastrointestinal administration may be carried out via nasogastric, nasoduodenal, nasojejunal tube, gastrostomy, or gastroscopic work channel [8,71].

The use of encapsulated stool preparations makes the administration of the transplant [81] easier and more tolerable. The mechanism of the excellent effect of FMT is still unclear today.

Based on a recent observational study by Péterfi and his team, it has been shown that the use of bacteria-free stool preparation in capsules is as effective as full-stool transplants [77,78,81,82] for the treatment of CDI.

### 5.3. Microbiological and Metabolic Changes after FMT

Bacteriophages are viruses that specifically target specific bacteria. Numerous studies have shown that the number of bacteriophages (*Caudovirales*) changes significantly, and that the success of transplantation is best when the amount of *Caudovirales* in the donor stool is higher [83].

Mycobiota changes: after successful transplantation, colonization occurs with different donor fungi belonging to the *Saccharomyces* and *Aspergillus* genera, while in unsuccessful transplantation, *Candida* species are dominant [8,83].

Metabonomics investigates metabolic responses due to various medications or other interventions. In the case of FMT, the significant role is played by short-chain fatty acids. In mice, it was found that the presence of high SCFA producers acted as a protective factor in inhibiting *C. difficile* growth. Valerate, butyrate, acetate, and propionate may also have an important protective effect in this condition and in the restoration of dysbiotic flora [83,84]. The results of some studies suggest that restoring intestinal microbial short-chain fatty acid producers via fecal microbiota transplantation (FMT) may result in the regulation of immunological reactions and the restoration of homeostatic equilibrium. At the same time, SCFAs can have a beneficial effect on the amount and function of regulatory T cells.

Immunological mechanisms suggest that the intestinal microbiome plays a significant role in the proper functioning of mucosal immunity, and through immunomodulation, it can also regulate the immune system of the whole organism. In addition to FMT, it has been observed that after transplantation, the antigen-presenting capacity of dendritic cells, monocytes, and MHC II-dependent macrophages decreases [82].

After FMT, Quraishi et al. observed a significant increase in IL-10-producing CD4 cells and a significant decrease in the IL-17-producing CD4 and CD8 cell populations. Overall, postbiotic products of FMT and intestinal microbiota can reduce inflammation by modulating regulatory T cells [85,86,87].

## 6. Discussion and Conclusions

The medication is still the primary form of care for migraines, but the previously mentioned treatments can be added on top of it. The fact that excessive medication use can hasten the course of this condition must be emphasized. Similar significance to managing migraines is played by the prevention strategies. A disease that is improperly prevented can advance. In a recently released research study, Katsuki and their coworkers illustrated the function of awareness campaigns in headache education and prevention [88].

As an alternative therapy, attempts to change the microbiome can be considered. The human intestinal microbiome contains the greatest number of bacteria in the organism and may influence metabolism [7]. It is an accepted fact that gut microbiota homeostasis is important for both the maintenance of the intestinal flora and brain physiology and normal function. The intestinal microbiome can become imbalanced through dietary changes, medication use, lifestyle choices, environmental factors, and aging [7]. The gut microbiota regulates neurotransmission, vascular barriers, cognition, and cerebral vascular physiology through their metabolites. The gut–brain axis may have an impact on migraine [1]. This interaction could be influenced by multiple factors such as inflammatory mediators, neuropeptides, serotonin pathways, stress hormones, and nutritional substances. The difficulties of metagenomic analysis for affected people and the interpretation of the fecal bacterial composition are obstacles to proper diagnosis and finding connections between bacteria and disease [28]. Another obstacle in current microbiota-based medicine is the difficulty in interpreting and tracking long-term effects. Specific microorganisms colonizing germ-free animals, probiotic use, and fecal microbiota transplantation approaches are widely used to elucidate the specific traits and functions of the microbiota [1].

Current evidence shows that the gut–brain axis influences migraines, even though the mechanism of this interaction is not fully clear. This interaction appears to be influenced by several factors, such as intestinal microbiota, neuropeptides, inflammatory mediators (IL-1β, IL-6, IL-8, and TNF-α), nutritional substances, stress hormones, and serotonin pathways. The neuropeptides, such as CGRP, SP, VIP, and NPY, are thought to have antimicrobial effects on various gut bacterial strains [1]. The relationship between the gut and the brain is bidirectional. In *H. pylori*-harboring patients, the pain might be improved after eradication therapy [23,24]. Conventional methods of modifying the microbiome, such as food, prebiotics, and probiotics, have shown some potential in animal models, and the results of human trials have been less compelling [7].

Dietary strategies can affect the course of migraines and are valuable tools for improving migraine management. However, due to limited evidence for the treatment of migraines with diet, a firm conclusion cannot be proved. It can be assumed that prescribing the diet is beneficial to the microbiota and the gut–brain axis. The dietary approaches include consumption of fiber, a low-glycemic index diet, vitamin D, probiotics, and weight-loss diet plans (for obese patients), which can also lead to an improvement in migraine-related characteristics [28].

A new, yet unproven, approach to treating migraines could be fecal transplantation. Gut microbial restoration is more effective and durable with FMT. The most important evidence for the efficacy and utility of FMT has been provided by randomized clinical trials. Currently, more than 340 clinical trials are underway to decide the place of FMT in clinical practice. The alteration of the gut microflora can lead to the normalization of inflammatory mediators and the serotonin pathway, which can influence the frequency and intensity of migraine pain. Randomized clinical trials are needed to prove the efficacy and utility of this method in the treatment of chronic migraines.

## 7. Strength of the Paper

The paper provides a comprehensive summary of the microbiological background of migraine. At the same time, the possibilities that can help to cure the disease by influencing the microbiome are presented. A detailed presentation of the mechanism of action of fecal transplantation may help in the development of targeted treatments at a later stage.

## 8. Weaknesses of the Paper

This paper could not show results using fecal transplantation; only hypothetical results are presented.

## Figures and Tables

**Table 1 microorganisms-11-02083-t001:** Metabolites involved in migraine pain reduction and microbes associated with the production of these metabolites.

Produced Metabolite	Connected Bacteria
Butyrate	*Faecalibacterium prausnitzii*, *Coprococus* spp., *Roseburia* spp., *Lachnosiraceae* spp., *Clostridial* Clusters IV and XIVa and *Eubacterium hallii*,
Propionate	*Bacteroides vulgatus*, *B. uniformis*, *Alistipes putredinis*, *Prevotella copri*, *Roseburia inulinivorans*, *Veilonella* spp., and *Akkermansia mucinophila*
GABA and acetylcholine	*Lactobacillus* spp. and *Bifidobacterium* ssp.
Serotonin	*Holdemania* spp., *Tyzzerella* spp., *Desulfovibrio* spp., *Yersinia* spp., *Bacillus* spp., *Clostridium* spp., *Ruminococcus* spp., *Enterococcus* spp., *Streptococcus* spp., and *Escherichia coli*
Tryptophan	*Actinobacteria* spp., *Firmicutes* spp., *Bacteroidetes* spp., *Proteobacteria* spp. and *Fusobacteria* spp.

**Table 2 microorganisms-11-02083-t002:** Intestinal phylums, genera, and species connected with migraine development. The positive effect means decreasing the migraine episode frequencies and intensity, while the negative effect means increasing the pain intensity or frequency.

Positive Effect on Migraine Development	Negative Effect on Migraine Development
*Akkermansia mucinophila*	*Clostridium coccoides*
*Alistipes putredinis*	*Clostridium propionicum*
*Bacteroides uniformis*	*Cytophaga hutchinsonii*
*Bacteroides vulgatus*	*Eggerthella lenta*
*Escherichia coli*	*Helicobacter pylori*
*Eubacterium hallii*	*Pseudomonas fluorescens*
*Faecalibacterium prausnitzii*	*Alcaligenes* spp.
*Prevotella copri*	*Burkholderia* spp.
*Roseburia inulinivorans*	*Cyanidium* spp.
*Ruminococcus gnavus*	*Fusobacteria* spp.
*Actinobacteria* spp.	*Pseudomonas* spp.
*Bifidobacterium* spp.	*Pseudonocardia* spp.
*Chloroflexi* spp.	*Ralsomia* spp.
*Clostridial* Clusters IV	*Rhodococcus* spp.
*Clostridial* Clusters XIVa	*Spirochetes* spp.
*Coprococcus* spp.	*Verrucomicrobiota* spp.
*Cyanobacteria* spp.	*Candida* spp.
*Desulfovibrio* spp.	*Micromycetes* spp.
*Enterococcus* spp.	*Streptpmyces* spp.
*Holdemania* spp.	
*Lachnosiraceae* spp.	
*Lactobacillus* spp.	
*Roseburia* spp.	
*Ruminococcus* spp.	
*Streptococcus* spp.	
*Tyzzerella* spp.	
*Veilonella* spp.	
*Yersinia* spp.	

**Table 3 microorganisms-11-02083-t003:** Summary of different possibilities for changing the gut microbiota and reducing the dysbiosis with their effects and disadvantages. Abbreviations: DASH—Dietary Approaches to Stop Hypertension; MIND—Mediterranean–DASH diet intervention for neurodegenerative delay; SCFA—short-chain fatty acid; NMDA—N-methyl-D-aspartate.

Options Used to Change the Gut Microbiota	Types and Mode of Action	Positive Effect	Disadvantage
Diet/prebiotics	Mediterranean diet, Japanese diet, MIND diet, DASH diet, Vegetarian diet, Ketogenic diet	Fewer migraine episodes, milder pain	Difficulty in maintenance, compliance problems, results after 3 months
Probiotics	SCFA production	Fewer migraine episodes, milder pain	No randomized human trials exist
Carbohydrates	Low-glycemic index carbohydrate increases SCFA level	Fewer migraine episodes, milder pain	Results after 3 months
Vitamin B2	Boosting SCFA level	Reduces number of migraine days by 1–3 days	Adjuvant therapy
Vitamin D3	Reduce *H. Pylori* quantity	Fewer migraine episodes, milder pain	Evaluable efficiency after 8 weeks
Magnesium	Affect NMDA receptor blockade	Fewer migraine episodes, milder pain	Adjuvant therapy
Eicosapentaenoic acid and docosahexaenoic acid	Antinociceptive effect	Milder pain	Adjuvant therapy
Fecal microbiota transplantation	Modification of gut microbiota, restore the microbiological balance	Theoretically reduces the migraine episodes and migraine severity	No randomized clinical trials

## Data Availability

No supplementary data was generated during the study.

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
