# Peer review of "Migraine as a Disease Associated with Dysbiosis and Possible Therapy with Fecal Microbiota Transplantation"

_microorganisms, 2023, doi:10.3390/microorganisms11082083_

Round 1

Reviewer 1 Report

The manuscript is a narrative review on migraine, microbiota, and fecal transplant. This is an interesting field, and FMT may be a potential therapeutic option for migraine. Overall, it is well-written but a bit hard to read.

General comments

1.     The title is misleading. There was insufficient discussion on the role of antibiotics in migraine pathogenesis. Is there any direct evidence supporting antibiotic use increases the incidence of migraine?

2.     Gut microbiota is likely associated with migraine pathophysiology (https://doi.org/10.1016/j.ynpai.2022.100090.) It would be nice to provide more direct/indirect evidence in animal studies or clinical trials to support such an association or causation.

3.     It is unclear how FMT benefits migraine or chronic pain. Is there any clinical evidence that restoration of immunological equilibrium improves headache or pain? There’s a neural link between the gut and the brain via the vagus nerve. Would you think it plays a role?

Specific comments

1.     Journal names were not abbreviated in many references. Ref#26 has no title.

2.     Line 129, “migraine sufferers …..vomited”. Please provide a reference.

No apparent grammar issue. Some topic sentences are not direct enough.

Author Response

Dear Reviewer!

I appreciate the feedback, the positive comments, and the suggestions for changes that would enhance our paper quallity. The requested modifications, which are listed below, have been made.

General comments

  1. The title is misleading. There was insufficient discussion on the role of antibiotics in migraine pathogenesis. Is there any direct evidence supporting antibiotic use increases the incidence of migraine?

Response 1: I appreciate your input. I must agree that not much is written regarding the connection between antibiotics and migraines. The main goal was to talk about how antibiotic treatment-induced dysbiosis and migraine are related. On lines 94–101 of the section "Causes of dysbiosis," we inserted a paragraph discussing the connection between antibiotics and migraines.

We propose changing the title to "Migraine as a disease associated with dysbiosis and possible therapy with faecal microbiota transplantation" if that is more acceptable.

  1. Gut microbiota is likely associated with migraine pathophysiology (https://doi.org/10.1016/j.ynpai.2022.100090.) It would be nice to provide more direct/indirect evidence in animal studies or clinical trials to support such an association or causation.

Response 2: I appreciate your suggestion. We attempt to give both direct and indirect evidence in animal studies and clinical trials supporting the relationship between gut microbiota and migraine in section "Role of gut microbiota in migraine," lines 130–143.

  1. It is unclear how FMT benefits migraine or chronic pain. Is there any clinical evidence that restoration of immunological equilibrium improves headache or pain? There’s a neural link between the gut and the brain via the vagus nerve. Would you think it plays a role?

Response 3. Regrettably, there is no clinical data linking FMT to migraines or chronic pain. The restoration of immunological equilibrium can improve headache or pain in animal models. The link between gut and brain via vagus nerve was introduce paper, section “Introduction” lines 58-60.

Specific comments

  1. Journal names were not abbreviated in many references. Ref#26 has no title.

Response 4. Updated references, journal name abbreviations, and Doi-s were all included.

  1. Line 129, “migraine sufferers …..vomited”. Please provide a reference.

Response 5. Reference was added at this section.

Reviewer 2 Report

Thanks for the detailedReview. Please specify that this is a Narrative Review, not a Systematic Review.

The authors narratively reviewed the association between migraine and microbiota. Such studies are still rare, and this summary is very important for organizing future research topics.

The relationship between intestinal bacteria and migraine is not well documented, at least in the Japanese guidelines for headache treatment, and I think this kind of review is very important to give many researchers a new perspective. It is a comprehensive review, properly stating that there are basic as well as epidemiological studies.

By summarizing the relationship between migraine and microbiota reported to date, we identify issues for future research (sample size and methodology).

Please mention that migraine can progress if not properly prevented, that medication overuse contributes to the progression process, and that this is why awareness campaigns and preventive therapies are important.

  • PMID: 36705435

Magnesium, vitamin B2, and EPA are also said to be associated with migraine, are they likely to be related to gut flora?

Author Response

Dear Reviewer!

I appreciate the feedback, the positive comments, and the suggestions for changes that would enhance our paper quality. The requested modifications, which are listed below, have been made.

The authors narratively reviewed the association between migraine and microbiota. Such studies are still rare, and this summary is very important for organizing future research topics.

The relationship between intestinal bacteria and migraine is not well documented, at least in the Japanese guidelines for headache treatment, and I think this kind of review is very important to give many researchers a new perspective. It is a comprehensive review, properly stating that there are basic as well as epidemiological studies.

By summarizing the relationship between migraine and microbiota reported to date, we identify issues for future research (sample size and methodology).

Thank you for your kind words of gratitude. I sincerely hope that the article's acceptance will assist other researchers in organising their fundamental research.

Please mention that migraine can progress if not properly prevented, that medication overuse contributes to the progression process, and that this is why awareness campaigns and preventive therapies are important.

Response 1.: The mentioned topic was introduced in paper on section „Discussion and conclusions” lines 334-340.

PMID: 36705435

Response 2. Thank you for making this advice; it was helpful in enhancing the paper's quality.

Magnesium, vitamin B2, and EPA are also said to be associated with migraine, are they likely to be related to gut flora?

Response 3.: I appreciate your thoughtful comment. Magnesium, Vitamin B2, and EPA are linked to migraines and have a connection to the flora of the gut. On lines 237–241 and 246-257 of the section "How to modify the gut microbiota," it is described the function of these chemicals in migraines and their relationship to gut flora.

Round 2

Reviewer 1 Report

This is a revision with a proposed title change. It was not seen in the revised manuscript. Otherwise, the authors have addressed most of my comments.

Still, I would recommend using subheadings, figures, or stronger topic sentences to improve the readability.

Author Response

Respected Reviewer,

Thank you for your helpful suggestions and comments once more.

This is a revision with a proposed title change. It was not seen in the revised manuscript. Otherwise, the authors have addressed most of my comments.

The title was changed after our previous proposal was accepted. The new title is"Migraine as a disease associated with dysbiosis and possible therapy with faecal microbiota transplantation”.

Still, I would recommend using subheadings, figures, or stronger topic sentences to improve the readability.

New subheadings and tables were added to the document.

You can view the changes on:

  1. lines 1 and 2. (title);
  2. lines 129, 133, 140, 145, 216, 232, 238, 244, 254, 267, 293, 307, 325, 326, 330, 330, 342 (subheadings)
  3. lines 185-192 and 285-290 (tables)

Thank you very much for your help and effort in improving the quality of this article.

Zoltan Peterfi MD